# Construction of model animals to explore intestinal microbiome for detection of breast cancer

Xia Ji[1☯]*, Ruipeng Li[1☯], Xiaoyu Hu[1], Yufang Tian[1], Liqiong Liu[1], Chenyu Zhang[1], Liangxiong Xu[1], Yongzhi Chen[1], Haiwei Xie[1], Lutian Mao[1], Tianshu Cai[2]*, Weiwei Li[2]*

**1** School of Life Science, Huizhou University, Huizhou, China, **2** Huizhou Health Sciences Polytechnic, Huizhou, China

☯ These authors contributed equally to this work.
* xiaji@hzu.edu.cn (XJ); liweiwei925@163.com (WL); caitianshu88@163.com (TC)

**Data Availability Statement:** *** PA @ ACCEPT: Please follow up with authors to confirm data available at Accept *** Our SRA records will be accessible with the following link after the indicated

## Abstract

Breast cancer ranks first among female cancers and has become a major public health problem in the current society. More studies indicated that these cancers are related to the change in the gut microbiome that can cause metabolic and immune system disorders in the body. However, there are few studies on the changes in gut microbiome caused by the onset of breast cancer, and the relationship between breast cancer and gut microbiome needs to be further clarified. In this study, we inoculated 4T1 breast cancer cells to induce breast cancer tumorigenesis in mice and collected their feces samples at different stages during this process. These intestinal florae were analyzed using 16S rRNA gene amplicon sequencing, and the results showed that at the phylum level, the ratio of *Firmicutes*/*Bacteroidetes* decreased with the development of the tumor; at the family level, the intestinal microbiome had obvious variations of *Lachnospiraceae, Bacteroidaceae, Erysipelotrichaceae*, etc. The Kyoto Encyclopedia of Genes and Genomes (KEGG) and COG annotation demonstrated that decreased abundance of cancer-related signaling pathways. This study elucidated the relationship between breast cancer and intestinal microbiome, and the research results can be used as an important biomarker for the diagnosis of breast cancer.

## Introduction

According to Global Cancer Statistics 2020 estimated that were about 2.3 million breast cancer incidences in the world in 2020, accounting for 11.7% of the total number of cancer cases, and has surpassed lung cancer, ranking first in the world cancer incidence [1]. In China, the incidence of breast cancer is rising every year, and now more than 300,000 women are diagnosed in 2020. With the improvement of medical care and health awareness, the death rate of breast cancer has decreased, but in rural and remote areas, the death rate of breast cancer was still high [2].

Research has shown that BRCA-1/2, p53 and PTEN genes mutation, unhealthy living habits, and obesity could increase the incidence of breast cancer [3–5]. However, in the early stage

release date: https://www.ncbi.nlm.nih.gov/sra/PRJNA892785.

**Funding:** This study was supported by grants from the Applied Basic Research Programs of Science and Technology Commission Foundation of Guangdong Province (No. 2022A1515012602) and Foundation of Guangdong Educational Committee for Youths (No. 2019KQNCX150). The funders had no role in study design, data collection and analysis, decision to publish, or preparation of the manuscript.

**Competing interests:** The authors have declared that no competing interests exist.

of breast cancer, the cure ratio of patients with active treatment can reach more than 90%, and there is no need for chemotherapy, radiotherapy, and other operations. But in the late stage, the cancer cells could spread through lymphatic metastasis, blood metastasis, implantation metastasis, direct metastasis, and other ways, and eventually appear as systemic multi-organ lesions, directly threatening the life of patients [6–9]. There are no effective measures to prevent breast cancer. Therefore, the early diagnosis of breast cancer plays an important role in the treatment and prognosis of patients.

The traditional diagnostic methods for breast cancer include breast self-examination, clinical physical examination, imaging examination, biopsy, and molecular biological examination [10, 11]. At present, no method can completely and accurately diagnose breast cancer early, which needs to be judged according to the actual situation of patients. Thus, the early diagnosis of breast cancer still needs to be supplemented by more effective methods. The gut microbiome is regarded as the "second genome" of humans, and the occurrence of many diseases will change the gut microbiome, such as diabetes, depression, cancer, and so on [12–14]. With the improvement of modern whole genome sequencing, the gut microbiome in the body can be efficiently detected [15]. There also has research shown that the occurrence of tumors is directly related to the change in the intestinal microbiome, which causes the metabolites disordered in the body, and even stimulates tumor-related signaling pathways, directly inducing the formation of tumor cells [16, 17]. So, biomarkers that alter gut microbiomes can be used as a standard to predict tumorigenesis or as an auxiliary means to diagnose tumorigenesis.

In this study, 4T1 breast cancer tumor cells were used to construct a mouse breast cancer model, and 16S rRNA sequencing was performed to analyze the intestinal microbiome of mice during tumorigenesis. The study of intestinal microbiome alternation during the tumorigenesis of breast cancer can provide a new direction for the prevention and early diagnosis and treatment of breast cancer.

## Materials and methods

### Ethics approval

All experiments involving animals were approved by the Animal Care and Use Committee of Huizhou University, China. Approval from the Science and Technology Agency of Guangdong Province, China, was obtained for all protocols (ID: SCXK (Yue) 2013-0003).

### Animal experiments

All experiments involving animals were approved by the Animal Care and Use Committee of Huizhou University and were performed according to the governmental guidelines of the Ministry of Science and Technology of the PR China for the Care and Use of Laboratory Animals. The 4T1 cells (ATCC, Manassas, VA) were cultured in Roswell Park Memorial Institute (RPMI) 1640 medium with 1% penicillin, 1% streptomycin, and 10% fetal bovine serum (FBS) at 37˚C under 5% $CO_2$. $6 \times 10^6$ of these cultured cells were collected and re-suspended in PBS. To obtain the breast cancer aminal model, these collected cells were inoculated into the flank of the specific pathogen-free (SPF) BALB/c mice (5 weeks old, 20–25 g, female). Then, these mice were divided equally into two groups and housed in pathogen-free conditions (22±2˚C, 50±5% humidity, 12 h light/dark cycle) with food and water freely to acclimatize for 7 days, tumor volume and body weight were measured every day and fecal samples were obtained. In our experiments, the maximum size of palpable tumor volume does not grow to more than 100 $mm^3$ and all the animals were sacrificed by 100 mg/kg pentobarbital administered via intraperitoneal injections. RPMI 1640, FBS, antibiotic-antimycotic solution, penicillin/

streptomycin, phosphate-buffered saline (PBS), and 0.25% (w/v) trypsin/ 1 mM EDTA were ordered from Gibco (Thermo Fisher Scientific, USA).

## 16S rRNA genes sequencing

Total DNA was extracted from the thawed colonic contents samples using the QIAamp DNA stool extraction kit (Qiagen, Germany) following the manufacturer's protocol. The genomic DNA was then examined by 1% agarose gel electrophoresis. All samples were quantified on a Qubit 2.0 Fluorometer (ThermoFisher Scientific, USA). Then, the V3-V4 hypervariable region of the 16S rRNA gene was amplified by PCR, which used 338F forward primer (5'-ACTCCTA CGGGAGGCAGCA-3') and the 806R reverse primer (5'-GGACTACHVGGGTWTCTAAT-3'). The PCR cycle was denaturation at 94˚C for 3min (1 cycle); followed by 94˚C for 45 s, annealing at 50˚C for 60 s, and extension at 72˚C for 90 s (25 cycles), and a final extension step of 72˚C for 10min. The amplicon products were purified with AMPure XP beads (Beckman Coulter, USA). Sequencing libraries were generated using the TruSeq® DNA PCR-Free Sample Preparation Kit (Illumina, USA) following the manufacturer's recommendations. The library quality was assessed using the Agilent Bioanalyzer 2100 system (Agilent Technologies, USA). The quality libraries were finally sequenced on an Illumina HiSeq 2500 platform with 250 bp of paired-end reads.

## Microbiome data analysis

The raw pair-end reads were overlapped and merged to get raw tags by FLASH software (v1.2.7) [18]. Trimmomatic (v0.33) is utilized to filter low-quality raw tags and kept high-quality clean tags [19]. Then, clean tags were imported into the software package Quantitative Insights Into Microbial Ecology 2 (QIIME2 version 2018-8) [20]. Within QIIME2, sequences were quality-filtered and denoised using the Divisive Amplicon Denoising Algorithm 2 (DADA2) pipeline [21]. Taxonomy was assigned using the 99% identity SILVA (release 132) V3-V4 classifier [22]. All the fecal microbiome ribosomal sequence variants (RSVs) were identified as unique features across all samples without clustering. R package ade4 ordinated in explanatory matrices using PCA including infection time course as explanatory variables. Alpha and beta diversity were computed using a rarefaction depth of sequences. Differences in bacterial alpha diversity (Shannon's index, observed features, and evenness index) between each study group were evaluated based on the rarefied data and tested by the Wilcox rank-sum method. The tool PICRUSt was applied to predict potential functional changes in the microbiome by inferring the metagenomes from 16S rRNA sequences [23].

## Statistical analyses

R (3.0.2; R Foundation for Statistical Computing) and Prism software (Graph Prism7.0 Software Inc. CA, USA) were used for statistical analysis. The results were expressed as a mean ± standard deviation (SD).

## Results

### Sample information

4T1 breast cancer cell line can spontaneously transform into highly metastatic cancer cells and spread to the whole body through lymphatic metastasis, blood metastasis, implantation metastasis, direct metastasis, and other ways. Thus, we used the mouse 4T1 cell line and BALB/c mice to establish the breast cancer model. Fecal samples were collected from mice before 4T1 cells inoculation (Day 0-1, Day 0-2), and then collected on Day 3 and Day 7 after 4T1 cells

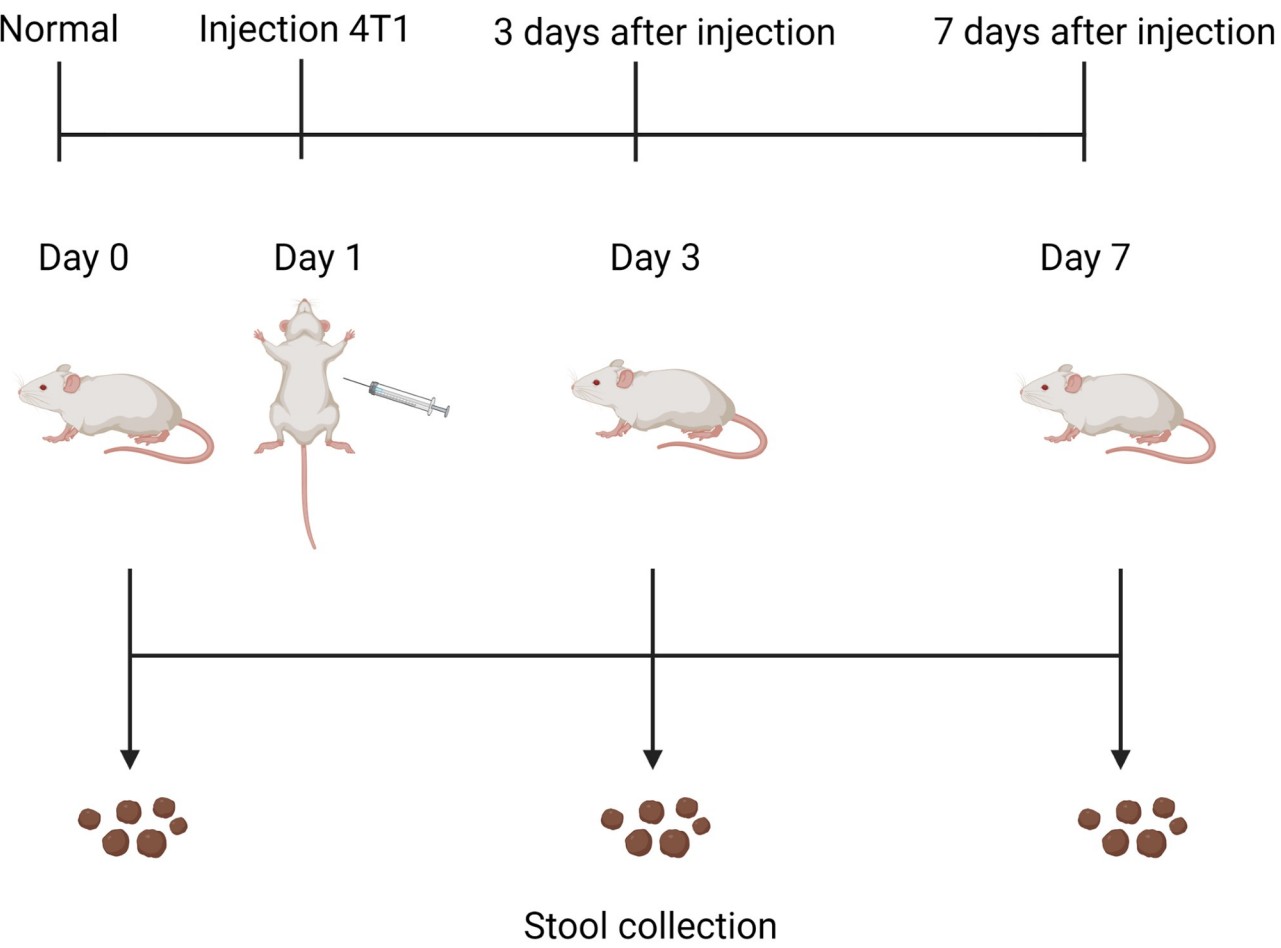

**Fig 1. Flow chart of animal experiment.**

inoculation (Day 3-1, Day 3-2, Day 7-1, Day 7-2). DNA extraction was performed on the collected fecal samples (Fig 1). The extracted fecal DNA was analyzed by 16S rRNA sequencing to determine the composition of the intestinal microbiome before and after tumorigenesis.

### The alpha diversity of the gut microbiome

To explore alteration in the microbiome community structure during the breast cancer pathogenesis of mice, the microbial alpha diversity was used to compare the microbiome at different time points during tumor formation. The results showed that, with the increase of tumorigenesis time, the alpha diversity of the gut microbiome in Day 3 and Day 7 mice was lower than that of Day 0 in terms of Evenness and Shannon index (Evenness index Day 0 = 0.91, Day 3 = 0.90, Day 7 = 0.89; Shannon Day 0 = 6.68, Day 3 = 6.50, Day 7 = 6.19) (Fig 2A and 2B).

### The beta diversity of the gut microbiome

The principal co-ordinates analysis (PCoA) by unweighted uniFrac distance and weighted uniFrac distance showed that the gut microbiome of breast cancer formed mice (Day 3 and Day 7) clustered significantly separately from that of breast cancer formed before mice (Day 0) (Fig 3).

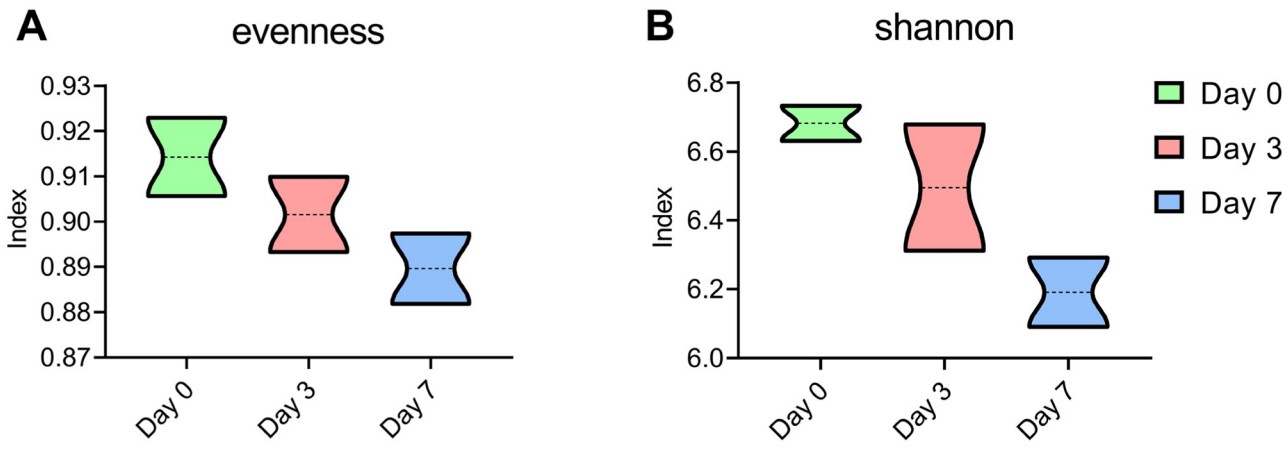

**Fig 2. Alpha diversity of intestinal microbiome in mice on different days.** (A) Evenness index. (B) Shannon index.

## Microbiome taxonomic abundance analysis of breast cancer mouse

To evaluate the gut microbial community features during the breast cancer tumorigenesis of mice, the relative taxonomic abundance of the microbiome in feces collection of mice before

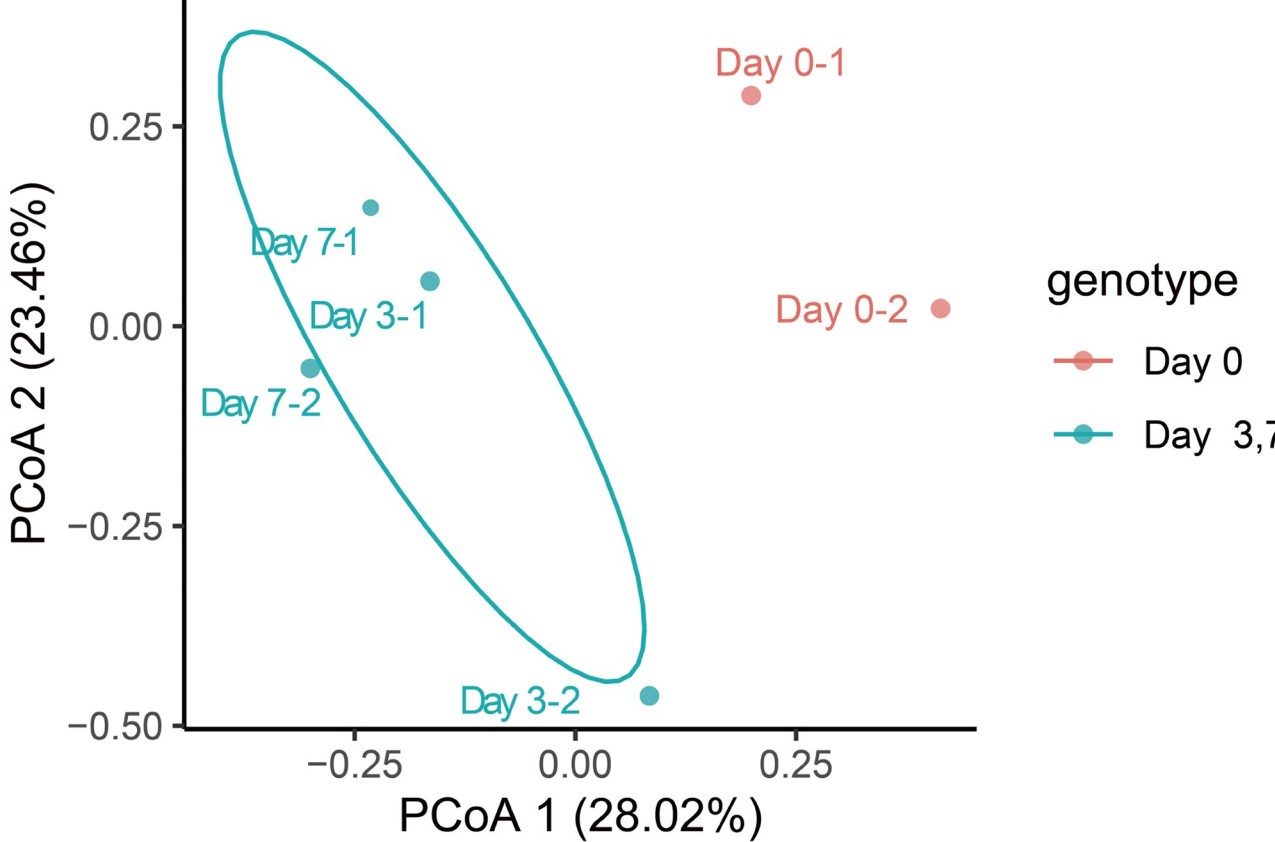

**Fig 3. PCoA (unweighted Unifrac distance) of the gut microbiome composition of mice before (Day 0), after tumorigenesis on the third day (Day 3) and on the seventh day (Day 7).**

**Table 1. Reads count of each sample.**

| Sample ID | Reads Count |
|---|---|
| Day 0-1 | 32844 |
| Day 0-2 | 33843 |
| Day 3-1 | 29482 |
| Day 3-2 | 35115 |
| Day 7-1 | 28054 |
| Day 7-2 | 28757 |

tumorigenesis, the third day, and the seventh day of tumorigenesis were compared. The total number of 250,664 reads were passed the quality control for all samples, and the analysis of operational taxonomic units (OTU) included 6 phyla and 25 families of all samples' gut microbes (Table 1). The Venn diagrams indicated that, with the formation of tumors, the phylum of the intestinal microbiome in mice did not have significant differences (Fig 4A). At the phylum level, *Firmicutes, Bacteroidetes, Tenericutes*, and *Actinobacteria* were the main dominant identified in all groups, contributing 98% of the gut bacteria (Fig 4B). With the tumorigenicity of mice, the abundance of *Bacteroidetes* and *Actinobacteria* gradually increased. The abundance of *Bacteroidetes* was 13.04% before tumorigenicity, and increased to 17.62% on the third day and 24.24% on the seventh day; that of *Actinobacteria* was from 0.86% to 1.81% on the third day and 12.99% on the seventh day. The ratio of *Firmicutes* to *Bacteroidetes* was 5.56 before tumorigenicity, 4.05 on the third day, and 2.93 on the seventh day. But the abundance of *Tenericutes* and *Proteobacteria* gradually decreased. The abundance of *Tenericutes* was 11.73% before tumorigenicity and decreased to 8.47% on the third day and 3.39% on the seventh day; that of *Proteobacteria* was from 1.15% before tumorigenicity to 0.13% on the third day and 0.08% on the seventh day (Fig 4B and 4C).

At the family level, there are differences in the composition of intestinal microbial communities in mice on different days of tumorigenesis, and the differences become more obvious with the increase of days (Fig 5A). With the tumorigenicity of mice. The abundance of *Lachnospiraceae* was 22.67% before tumorigenicity and increased to 25.38% on the third day and 27.66% on the seventh day (Figs 5B, 5C and 6A); that of *Bacteroidaceae* was from 12.19% to 16.50% on the third day and 22.58% on the seventh day (Figs 5B, 5C and 6B); *Erysipelotrichaceae* increased from 2.31% to 5.62% and 6.16% (Figs 5B, 5C and 6C). But the abundance of *Ruminococcaceae* gradually decreased that was from 40.70% before tumorigenicity to 34.48% on the third day and 30.26% on the seventh day (Figs 5B, 5C and 6D); that of *f_Mollicutes_RF39_uncultured_bacterium* was from 6.41% to 2.63% on the third day and 0.95% on the seventh day (Figs 5B, 5C and 6E); *o_Mollicutes_RF39* decreased from 4.15% to 3.55% and 0.88% (Figs 5B, 5C and 6F). And *Staphylococcaceae* was present only before tumor formation; *Eubacteriaceae* was present only after tumor formation (Figs 5B, 5C, 6G and 6H).

## Functional profile of the gut microbiome

KEGG and COG pathway analyses were used to compare the functional composition of the gut microbiome in mice before (Day 0) and after (Day 7) tumor growth. The results showed that after tumorigenesis mouse intestinal microbiome abundance of the pathways in energy metabolism, amino acid metabolism, carbohydrate metabolism, lipid metabolism, metabolism of cofactors and vitamins, cell motility, nucleotide metabolism, translation, transcription, signal transduction, replication and repair, and membrane transport were not as high as before the tumor (Fig 7A and 7B).

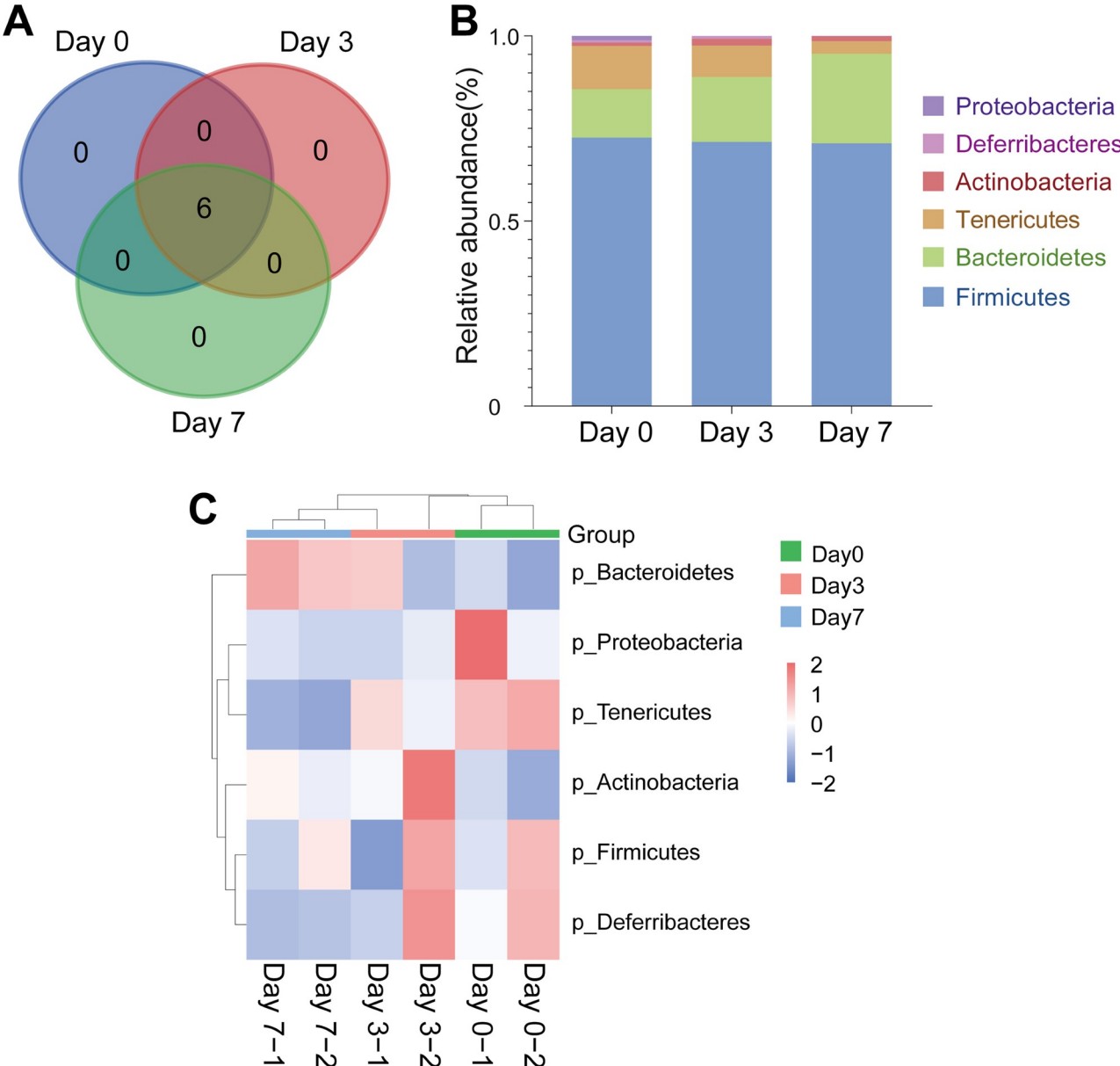

**Fig 4.** (A) Venn diagram of the gut microbial community at the phylum level on different days of tumorigenesis in mice. (B) Histogram of the gut microbial community at the phylum level of different groups. (C) Heat map of the gut microbial community at the phylum level of different groups.

## Discussion

In this study, 4T1 breast tumor cells were inoculated in mice to build a breast cancer tumor model, and 16S rRNA sequencing was used to analyze the changes in intestinal flora in the tumor-forming process. The results showed that the abundance of the intestinal microbiome of the mouse after tumorigenesis was decreased.

We found that with the growth of the tumor, there were significant differences in *Tenericutes, Bacteroidetes, Actinobacteria*, and *Proteobacteria* at the phylum level. *Firmicutes* and *Bacteroidetes* are the dominant bacteria of intestinal microbial community structure [24, 25], and the ratio of F/B (the ratio of *Firmicutes* and *Bacteroidetes*) is significantly lower in cancer

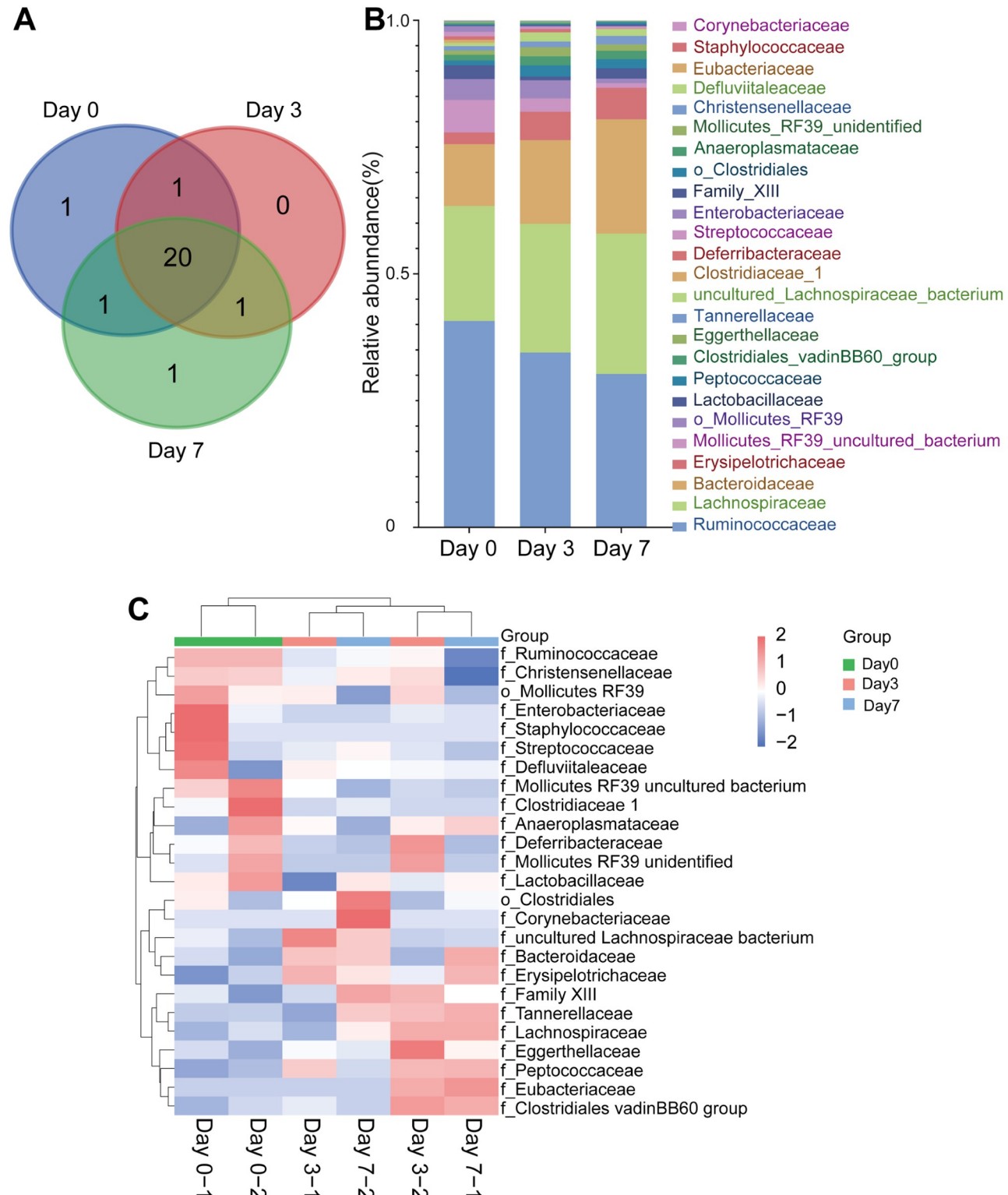

**Fig 5.** (A) Venn diagram of the gut microbial community at the family level on different days of tumorigenesis in mice. (B) Histogram of the gut microbial community at the family level of different groups. (C) Heat map of the gut microbial community at the family level of different groups.

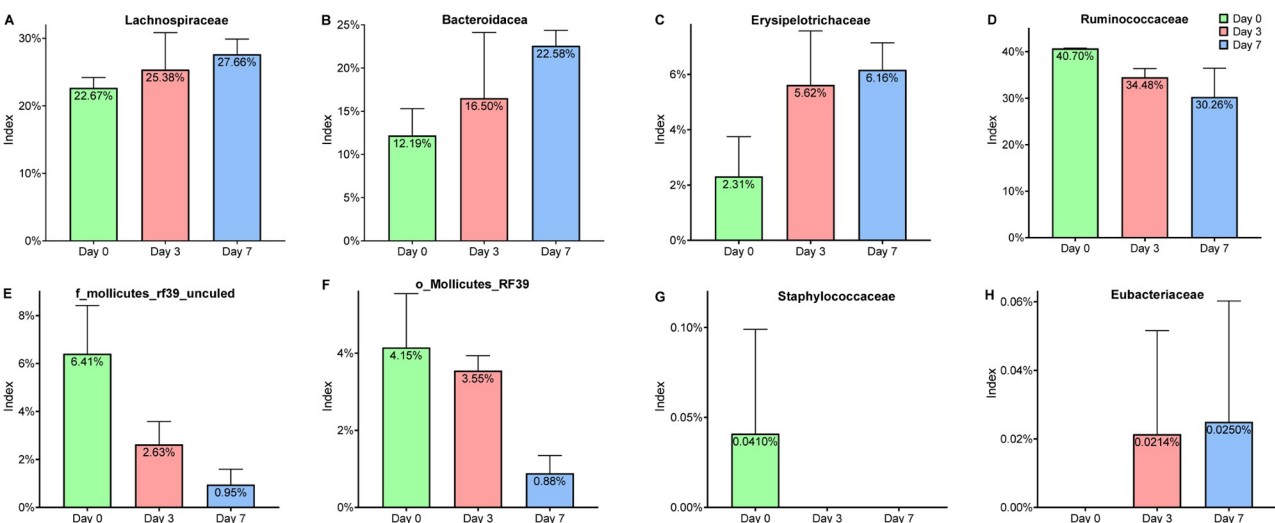

**Fig 6. The taxonomic summary demonstrated the OTUs are assigned to the prevalent microbiome of *Lachnospiraceae* (A), *Bacteroidacea* (B), *Erysipelotrichaceae* (C), *Ruminococcaceae* (D), *f_Mollicutes_RF39_uncultured_bacterium* (E), *o_Mollicutes_RF39* (F), *Staphylococcaceae* (G) and *Eubacteriaceae* (H) at the family level of different stages.**

patients than in healthy people [26]. Our results showed that the F/B value of intestinal micro-organisms in mice decreased with the longer time of tumor growth. Sheetal Parida also found similar results in the gut microbiome of breast cancer [27]. *Firmicutes* in the gut flora ferments carbohydrates into short-chain fatty acids. When the abundance of *Firmicutes* decreases, the synthesis of these short-chain fatty acids decreases, which can lead to reduced intestinal function [28, 29].

At the family level, we found that after inoculating tumors in mice, the relative abundance ratio of *Erysipelotrichaceae, Bacteroidaceae*, and *Lachnospiraceae* increased obviously, and the relative abundance of *Ruminococcaceae* reduced. All of these bacteria have been associated with cancer: in colon cancer, *Erysipelotrichaceae* were higher than in healthy people [30]; dys-regulation of *Bacteroidaceae* can lead to peritonitis, intra-peritoneal abscess, and bacteremia [31]; the relative abundance of *Lachnospiraceae* also varied significantly in individuals with cancer such as lung cancer and colon cancer [32, 33]; and *Ruminococaceae* decreases during breast cancer tumor development, and cause mucosal inflammation, impaired barrier function and reduced intestinal permeability [34].

As tumors proliferate, metabolic and immune pathways in the body are affected, and these can be used as indicators of cancer diagnosis [35, 36]. The analysis of COG and KEGG showed that changes in the gut microbiome could lead to a metabolic decline in mice, which is similar to the data of cancer diagnosis, such as carbohydrate metabolism, amino acid metabolism, energy metabolism, nucleotide metabolism, and lipid metabolism. This is consistent with our previous results of microbiome taxonomic abundance analysis. The decrease in the F/B ratio is the main reason for the low concentration of circulating short-chain fatty acids in carbohy-drate metabolism. *Lachnospiraceae* and *Ruminococcaceae* are related to the production and uptake of butyrate in carbohydrate metabolism, which regulates intestinal immunity [37]. *Bac-teroidaceae* mediated carcinogenesis through *β*-catenin and Notch1 pathways [17]. The growth of the tumor will affect the balance of intestinal microbiome, leading to the dysfunction of intestinal microbiome, and then aggravating the growth of the tumor. The variation in the abundance of bacterial species found in this study is slightly different from the results of other studies, which may be caused by the differences in the environment of human clinical samples

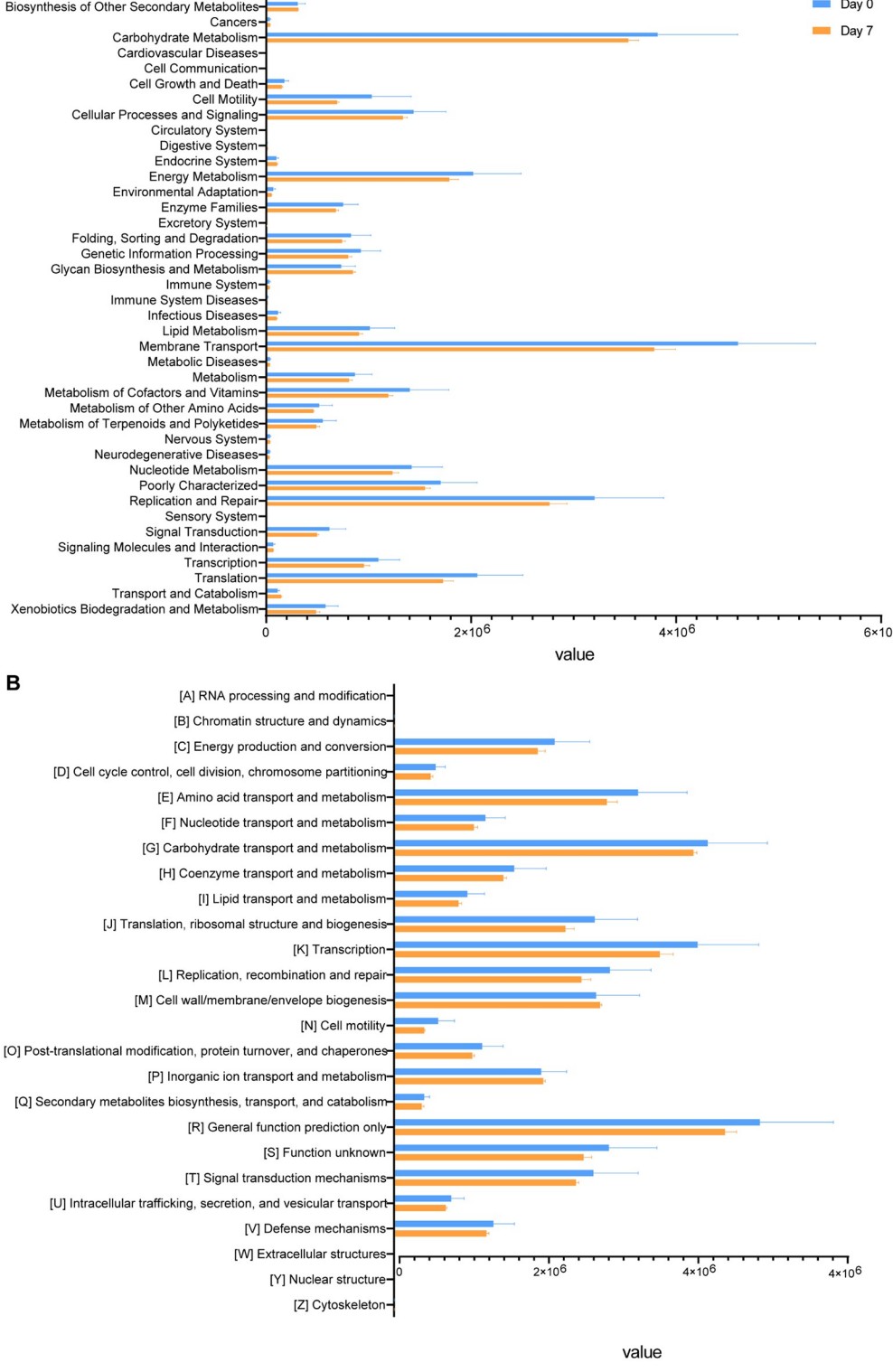

**Fig 7.** (A) KEGG pathway was less abundant after tumor formation than before tumor formation. (B) COG pathway was less abundant after tumor formation than before tumor formation.

or the host of different species. Laboratory mouse models with the same culture environment can overcome this, and if the clinical samples are combined with this project, it will have better practical significance.

In summary, this study demonstrates that breast cancer tumor formation can lead to an imbalance of intestinal microecology in the mouse. This can be used as an indicator for the detection or auxiliary diagnosis of breast cancer and provide a new idea for the staging diagnosis of breast cancer.

## Acknowledgments

The authors thank Natural Products Laboratory, Huizhou University, for the technical assistance and generous leading of time and equipment.

## Author Contributions

**Writing – original draft:** Ruipeng Li, Xiaoyu Hu, Yufang Tian, Liqiong Liu, Chenyu Zhang, Haiwei Xie, Lutian Mao, Tianshu Cai, Weiwei Li.

**Writing – review & editing:** Xia Ji, Liangxiong Xu, Yongzhi Chen, Tianshu Cai, Weiwei Li.

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
