## [Decision Letter · Decision Letter 0]

21 Feb 2023

PONE-D-23-00835Construction of Model Animals to Explore Intestinal Microbiome for Detection of Breast CancerPLOS ONE

Dear Dr. Ji,

Thank you for submitting your manuscript to PLOS ONE. After careful consideration, we feel that it has merit but does not fully meet PLOS ONE’s publication criteria as it currently stands. Therefore, we invite you to submit a revised version of the manuscript that addresses the points raised during the review process.

We look forward to receiving your revised manuscript.

Kind regards,

MUHAMMAD SHAHID RIAZ RAJOKA, Ph.D.

Academic Editor

PLOS ONE

Journal Requirements:

2. At this time, we request that you  please report additional details in your Methods section regarding animal care, as per our editorial guidelines: 

1) Please provide details of animal welfare (e.g., shelter, food, water, environmental enrichment) 

2) Please describe any steps taken to minimize animal suffering and distress, such as by administering anesthetics or analgesics

3) Please include the maximum sizes tumors grew to in your experiments and 

4) Please describe the post-operative and post tumor implantation care received by the animals, including the frequency of monitoring and the criteria used to assess animal health and well-being. Thank you for your attention to these requests.

"Xia JI, Weiwei LI and Tianshu Cai designed this study. Xia Ji, Ruipeng Li, Xiaoyu Hu, Yufang Tian, Liqun Liu and Chenyu Zhang performed experiments, analyzed data and interpreted results. All authors wrote and approved the manuscript."

"This study was supported by grants from the Applied Basic Research Programs of Science and Technology Commission Foundation of Guangdong Province (No. 2022A1515012602), Foundation of Guangdong Educational Committee for Youths (No.2019KQNCX150), and Huizhou University Supporting Foundation for PhD (No.2019JB038)."

"Xia JI, Weiwei LI and Tianshu Cai designed this study. Xia Ji, Ruipeng Li, Xiaoyu Hu, Yufang Tian, Liqun Liu and Chenyu Zhang performed experiments, analyzed data and interpreted results. All authors wrote and approved the manuscript."

6. Thank you for stating the following in your Competing Interests section:  

"NO authors have competing interests."

7. We note that you have stated that you will provide repository information for your data at acceptance. Should your manuscript be accepted for publication, we will hold it until you provide the relevant accession numbers or DOIs necessary to access your data. If you wish to make changes to your Data Availability statement, please describe these changes in your cover letter and we will update your Data Availability statement to reflect the information you provide.

8. PLOS requires an ORCID iD for the corresponding author in Editorial Manager on papers submitted after December 6th, 2016. Please ensure that you have an ORCID iD and that it is validated in Editorial Manager. To do this, go to ‘Update my Information’ (in the upper left-hand corner of the main menu), and click on the Fetch/Validate link next to the ORCID field. This will take you to the ORCID site and allow you to create a new iD or authenticate a pre-existing iD in Editorial Manager. Please see the following video for instructions on linking an ORCID iD to your Editorial Manager account: https://www.youtube.com/watch?v=_xcclfuvtxQ

9. Please amend the manuscript submission data (via Edit Submission) to include authors: Tianshu Cai, Weiwei Li

Reviewers' comments:

Reviewer's Responses to Questions

**Comments to the Author**

1. Is the manuscript technically sound, and do the data support the conclusions?

Reviewer #1: Yes

Reviewer #2: Yes

2. Has the statistical analysis been performed appropriately and rigorously? 

Reviewer #1: Yes

Reviewer #2: Yes

3. Have the authors made all data underlying the findings in their manuscript fully available?

Reviewer #1: Yes

Reviewer #2: Yes

4. Is the manuscript presented in an intelligible fashion and written in standard English?

Reviewer #1: Yes

Reviewer #2: Yes

5. Review Comments to the Author

Reviewer #1: Reviewer Comments:

This is a well written manuscript, thanks to author. The authors adequately represent the most relevant and recent advances in the field. The objectives and the rationale of this study are clearly stated. The methods, statistical analyses, controls, sampling mechanism, and statistical reporting (e.g.,Mean +/- SD) are appropriate, rigorous, and well described. The interpretations of results and study conclusions are supported by the data. The authors have clearly emphasized the strengths of their study, but they have not clearly stated the limitations of their study.

This Manuscript should be accepted for publication.

Reviewer #2: Check grammer and discribe more about the breast cancer model and methods for generating the model for breast cancer studies.

Also explain about the way you characterise various microbes in the gut.

Over all mauscript is well written and I will acceptance of paper after minor corrections.

6. PLOS authors have the option to publish the peer review history of their article (what does this mean?). If published, this will include your full peer review and any attached files.

Reviewer #1: No

Reviewer #2: No

While revising your submission, please upload your figure files to the Preflight Analysis and Conversion Engine (PACE) digital diagnostic tool, https://pacev2.apexcovantage.com/. PACE helps ensure that figures meet PLOS requirements. To use PACE, you must first register as a user. Registration is free. Then, login and navigate to the UPLOAD tab, where you will find detailed instructions on how to use the tool. If you encounter any issues or have any questions when using PACE, please email PLOS at figures@plos.org. Please note that Supporting Information files do not need this step.<quillbot-extension-portal></quillbot-extension-portal>

---

## [Author Response · Author response to Decision Letter 0]

10 Apr 2023

To the academic editor:

1. We have revised our manuscript to PLOS ONE's style requirements and highlight them in papers.

2. We report additional details in your Methods section regarding animal care, as the editorial guidelines at this time: 

1) We provide details of animal welfare including shelter, food, water, and environmental enrichment in line 98.

2) We used 100 mg/kg pentobarbital to administer intraperitoneal injections to euthanasia of the mice that minimize animal suffering and distress in line 102.

3) In our experiments, the maximum size of tumors does not grow to more than 100 mm3, and we provide this in line 102.

4) We describe the post-operative and post tumor implantation care received by the animals, including the frequency of monitoring and the criteria used to assess animal health and well-being in line 100.

3. We have corrected the grant information in the “Funding Information” and “Financial Disclosure”.

4. We state that “This study was supported by grants from the Applied Basic Research Programs of Science and Technology Commission Foundation of Guangdong Province (No. 2022A1515012602) and Foundation of Guangdong Educational Committee for Youths (No. 2019KQNCX150). The funders had no role in study design, data collection and analysis, decision to publish, or preparation of the manuscript.” And we include this amended Role of Funder statement in our cover letter.

5. We have deleted funding information in the Acknowledgments section and include your amended statements within your cover letter.

6. We have completed our Competing Interests on the online submission form to state any Competing Interests as "The authors have declared that no competing interests exist." And we include this amended Role of Funder statement in our cover letter.

7. We have changed our Data Availability provide information statement to be released, and the Bioproject was released and include your amended statements within your cover letter.

8. I ensure that I have an ORCID Id (0009-0008-2283-1733) and that it is validated in Editorial Manager. 

9. We have not found the institution and department of Tianshu Cai and Weiwei Li in the submission part, could you please add their information? Their institution is Huizhou Health Sciences Polytechnic and the department is Department of Pharmacy and Laboratory.

10. We have reviewed our reference list to ensure that it is complete and correct, and we have not cited papers that have been retracted.

To reviewer1:

1. We considered that the limitations of their study were that the variation in the abundance of bacterial species found in this study is slightly different from the results of other studies, which may be caused by the differences in the environment of human clinical samples or the host of different species. Laboratory mouse models with the same culture environment can overcome this, and if the clinical samples are combined with this project, it will have better practical significance., and we add this in the discussion in line 254.

To reviewer2:

1. We have checked grammar in the manuscript again.

2. We have described more about the breast cancer model and methods for generating the model for breast cancer studies in line 95.

3. We also added more about the way our characterize various microbes in the gut in line 227.

---

## [Editor Report · Decision Letter 1]

27 Apr 2023

Construction of Model Animals to Explore Intestinal Microbiome for Detection of Breast Cancer

PONE-D-23-00835R1

Dear Dr. Ji,

We’re pleased to inform you that your manuscript has been judged scientifically suitable for publication and will be formally accepted for publication once it meets all outstanding technical requirements.

Kind regards,

MUHAMMAD SHAHID RIAZ RAJOKA, Ph.D.

Academic Editor

PLOS ONE

Additional Editor Comments (optional):

Reviewers' comments:

<quillbot-extension-portal></quillbot-extension-portal>

---

## [Editor Report · Acceptance letter]

8 May 2023

PONE-D-23-00835R1 

Construction of Model Animals to Explore Intestinal Microbiome for Detection of Breast Cancer 

Dear Dr. Ji:

I'm pleased to inform you that your manuscript has been deemed suitable for publication in PLOS ONE. Congratulations! Your manuscript is now with our production department. 

Kind regards, 

on behalf of

Dr. MUHAMMAD SHAHID RIAZ RAJOKA 

Academic Editor

PLOS ONE